# FT-GAN: Face Transformation with Key Points Alignment for Pose-Invariant Face Recognition

**Weiwei Zhuang [1], Liang Chen [2,*,†], Chaoqun Hong [1], Yuxin Liang [1] and Keshou Wu [1]**

[1]  School of Computer and Information Engineering, Xiamen University of Technology, Xiamen 361024, China
[2]  School of Data and Computer Science, Sun Yat-Sen University, Guangzhou 510006, China
*  Correspondence: chenliang6@mail.sysu.edu.cn
†  Current address: East Campus, Sun Yat-sen University, Guangzhou 361024, Guangdong, China.

**Abstract:** Face recognition has been comprehensively studied. However, face recognition in the wild still suffers from unconstrained face directions. Frontal face synthesis is a popular solution, but some facial features are missed after synthesis. This paper presents a novel method for pose-invariant face recognition. It is based on face transformation with key points alignment based on generative adversarial networks (FT-GAN). In this method, we introduce CycleGAN for pixel transformation to achieve coarse face transformation results, and these results are refined by key point alignment. In this way, frontal face synthesis is modeled as a two-task process. The results of comprehensive experiments show the effectiveness of FT-GAN.

**Keywords:** face recognition; face pose transformation; generative adversarial networks; key points alignment

## 1. Introduction

Face recognition is one of the most successful applications of computer vision, and has been widely used in personal identification, video surveillance, mobile payment, and more. There are also several companies (e.g., Face++, SenseTime) promoting face recognition products. However, current face recognition methods usually require frontal faces with ideal lighting condition. Therefore, the applications are still limited. For example, for scenarios where the users are aware of face capturing (e.g., personal identification and mobile payment), good performance can be achieved. On the other hand, if faces are captured without users' awareness, poses are complex. In this way, the performance is significantly reduced. Therefore, face recognition in the wild has become a hot topic [1].

Many researchers have tried to to solve the problem of complex face poses. The straightforward idea is to use multi-view face images [2]. Different models have been proposed to explore the correlation among views. In this way, the robustness of face matching is improved. Li et al. proposed canonical correlation analysis (CCA) to learn the multi-view subspace which combines different views [3]. Furthermore, Akaho extended CCA to kernel CCA (KCCA) and proposed a non-linear model [4]. With KCCA, Wang et al. achieved better performance on the face data by introducing the deeply coupled auto-encoder network (DCAN) method to make better use of the label information [5]. A similar idea was used by Sharma et al. [6]. They extended the generalized multiview linear discriminant analysis (GMLDA) approach to kernel GMLDA (KGMLDA).

However, multi-view images are not always available. Therefore, pose-invariant features are proposed. To compute pose-invariant features, facial landmarks are extracted. For example, Chen et al. extracted multi-scale local binary pattern (LBP) features from 27 landmarks and concatenated them to form a feature vector [7]. However, simple concatenation brings about highly non-linear intra-personal variation. To tackle this problem, Ding et al. combined component-level and landmark-level methods to obtain pose-robust features [8]. Due to the complex facial poses, landmark detection in unconstrained

images is difficult, and the performance is influenced. To solve this problem, authors in [9] extracted facial features around facial keypoints without landmark detection. Deep learning has also been used. Yim et al. interconnected two deep neural networks (DNNs) and extracted the pose-invariant features. A multi-task strategy was used to ensure that these features could be used to reconstruct the face image under the target pose and recover the original input face image.

Recently, frontal face synthesis based on 2D images has attracted a great deal of attention, which attempts to recover fontal faces with unconstrained poses. Neural networks are widely used for this purpose. Zhang et al. employed auto-encoders with only a single hidden layer for face frontalization [10]. Kan et al. utilized stacked auto-encoders to synthesize frontal faces from non-frontal faces with a progressive method. In this way, the difficulty of face synthesis for each auto-encoder is reduced [11]. In the past three years, GAN (generative adversarial network)-based methods have become increasingly popular. Some methods tried to compute 3D face shapes. For example, Hassner et al. explored a simple approach to approximate the 3D surface of 2D images [12]. Yin et al. proposed a deep 3D morphable model (3DMM) conditioned face frontalization generative adversarial network (FF-GAN) to generate frontal face images [13]. On the other hand, some methods learn the complete information without using 3D shapes. Hong et al. used a deep architecture with domain-adversarial training to combine domain adaptation, feature extraction, and classification for face recognition with a single sample per person [14]. Many researchers used the multi-task idea to improve the performance of networks. Tian et al. used an encoder–decoder network followed by a discriminator network to learn the complete representations with a generative adversarial network (CR-GAN) [15]. Huang et al. proposed a two-pathway generative adversarial network (TP-GAN) to synthesize frontal faces by simultaneously perceiving global and local information. Tran et al. jointly performed face frontalization and learned a pose-invariant representation to leverage each other. In this way, the disentangled representation learning-generative adversarial network (DR-GAN) was proposed [16]. State-of-the-art techniques in GAN such as cycleGAN are able to learn the mapping from one image domain to another [17].

Although frontal face synthesis has been widely studied, how to retain the features of the original faces is still a challenging problem. CycleGAN has shown effectiveness in mapping from one image domain to another. Simple pixel translation cannot provide reasonable results in frontal face transformation such as complex poses and part occlusion. To tackle this problem, we propose face transformation with key points alignment based on generative adversarial networks, namely, FT-GAN. In contrast to conventional CycleGAN, frontal face transformation with FT-GAN is modeled as a two-task process consisting of pixel-to-pixel transformation and key point alignment. In this way, the transformed images are refined by the positions of key points. The key contributions of FT-GAN are summarized below:

1. First, CycleGAN is introduced to frontal face transformation. The mapping relationship from unconstrained faces to frontal faces is computed by pixel-to-pixel transformation.
2. Second, key point alignment in introduced to refine the transformed results of CycleGAN. The positions of key points are extracted, and their spatial relationship is retained.
3. Finally, based on FT-GAN, pose-invariant face recognition can be achieved. We conducted comprehensive experiments on two challenging benchmark datasets to validate the effectiveness of the proposed method.

The remainder of this paper is organized as follows. The proposed FT-GAN for pose-invariant face recognition is presented in Section 2. After that, we demonstrate the effectiveness of FT-GAN on frontal face transformation and face recognition with complex poses. To show this, experimental comparisons were conducted with other state-of-the-art methods, and the results are presented in Section 3. We conclude in Section 4.

## 2. FT-GAN for Pose-Invariant Face Recognition

### 2.1. Overview of FT-GAN

The flowchart of the proposed framework is shown in Figure 1. The process of FT-GAN is modeled as a two-task learning. The first task is pixel-to-pixel transformation. CycleGAN is employed to synthesize a coarse frontal face. The second task is key point alignment. The above two tasks are conducted simultaneously within the GAN framework. With the facial key points, we can refine the transformed face by keeping the spatial relationships of facial landmarks. Finally, an improved result of frontal face transformation is achieved.

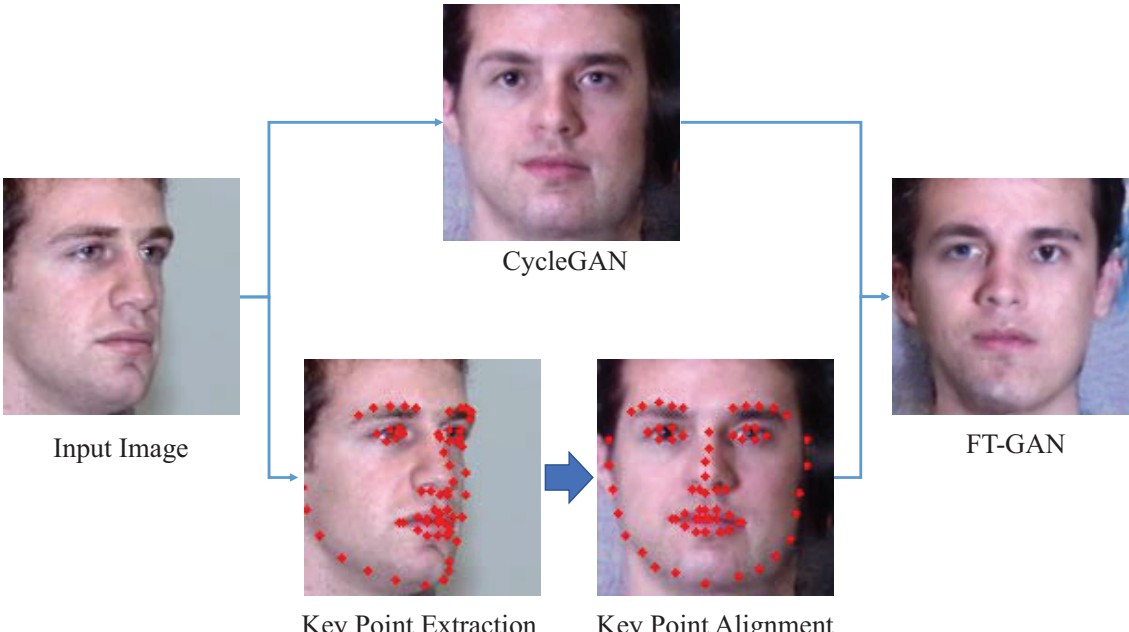

**Figure 1.** Flowchart of the proposed framework. Without loss of generality, we use gray images in the example. For each input image, we use CycleGAN to synthesize a frontal face. Besides, facial key points are extracted and used to align the facial landmarks. In this way, the result of CycleGAN is refined and an improved result is computed.

### 2.2. Face Transformation with Key Points Alignment

The goal of face transformation is to learn mapping function from the domain of unconstrained poses $X$ to the domain of frontal poses $Y$ giving training samples $x_{i_{i=1}}^{M}$ and $y_{i_{i=1}}^{N}$. $M$ and $N$ are the number of unconstrained faces and frontal faces, respectively. Note that they are not always the same since a frontal face may correspond to multiple unconstrained faces.

As mentioned before, CycleGAN has shown the ability of pixel-to-pixel transformation from one domain to another. Therefore, we use it in frontal face synthesis. Besides, due to the limitation of simple pixel translation, we extract key points of faces and use them to impose spatial constraints. In this way, the two tasks of FT-GAN are pixel-to-pixel transformation and key point alignment. Then, the objective function of FT-GAN can be defined by:

$$\mathcal{L}_{FT-GAN} = \mathcal{L}_{Pixel} + \lambda_1 \mathcal{L}_{KP}, \tag{1}$$

where $\mathcal{L}_{Pixel}$ is the loss of pixel-to-pixel transformation and $\mathcal{L}_{KP}$ is the loss of key point alignment. The trade-off between the two losses is denoted by parameter $\lambda_1$.

Suppose that $G$ represents the pose mapping function from $X$ to $Y$ and $\overline{G}$ represents the pose mapping function from $Y$ to $X$. $D_X$ and $D_Y$ are the pose mapping discriminators for $X$ and $Y$, respectively. The objective function of pixel-to-pixel pose transformation learning can be formulated as:

$$
\begin{aligned}
\mathcal{L}_{Pixel} = \ &\mathcal{L}_{GAN}(G, D_Y, X, Y) \\
&+\mathcal{L}_{GAN}(\overline{G}, D_X, Y, X) \\
&+\lambda_2 \mathcal{L}_{cyc}(G, \overline{G}),
\end{aligned}
\tag{2}
$$

where $\mathcal{L}_{GAN}$ represents the standard adversarial loss [18], $\mathcal{L}_{cyc}$ represents the cycle consistency loss [19], and $\lambda_2$ is the balanced parameter to control the trade-off between $\mathcal{L}_{GAN}$ and $\mathcal{L}_{cyc}$. Integrating Equations (1) and (2), we can obtain:

$$
\begin{aligned}
\mathcal{L}_{FT-GAN} = \ &\mathcal{L}_{GAN}(G, D_Y, X, Y) \\
&+\mathcal{L}_{GAN}(\overline{G}, D_X, Y, X) \\
&+\lambda_2 \mathcal{L}_{cyc}(G, \overline{G}) \\
&+\lambda_1 \mathcal{L}_{KP}(G, \overline{G}).
\end{aligned}
\tag{3}
$$

In the above equation, $\mathcal{L}_{GAN}$ is defined as:

$$
\begin{aligned}
\mathcal{L}_{GAN}(G, D_Y, X, Y) = \ &\mathbb{E}_{x \sim p_{data}(x)}[log D_Y(y)] \\
&+\mathbb{E}_{y \sim p_{data}(y)}[1 - D_Y(G(X))],
\end{aligned}
\tag{4}
$$

where $G$ aims at synthesizing images $G(x)$ similar to images in domain $Y$, while $D_Y$ aims at distinguishing between transformed samples $G(x)$ and real samples $y$. $\mathcal{L}_{GAN}$ is the key loss of GAN. Theoretically speaking, $G$ and $\overline{G}$ can be trained to produce outputs with identical distribution to target domains $Y$ and $X$ respectively.

However, if the size of the data is large enough, a network can map the same $x_i$ to any random permutation of images in $Y$, where any of the trained mappings can induce an output distribution that matches the target distribution. Thus, using $\mathcal{L}_{GAN}$ only cannot ensure that the learned mapping function can transform an input $x_i$ to a desired output $y_i$. To solve this problem, the cycle consistency loss $\mathcal{L}_{cyc}$ is proposed. For each image $x$ in $X$, the image transformation cycle should be able to bring $x$ back to origin, which can be described as:

$$
\begin{cases}
x \to G(x) \to \overline{G}(G(x)) \approx x, \\
y \to \overline{G}(y) \to G(\overline{G}(y)) \approx y.
\end{cases}
\tag{5}
$$

In this way, $\mathcal{L}_{cyc}$ is defined as:

$$
\begin{aligned}
\mathcal{L}_{cyc} = \ &\mathbb{E}_{x \sim p_{data}(x)}[\overline{G}(G(x)) - x] \\
&+\mathbb{E}_{y \sim p_{data}(y)}[G(\overline{G}(y)) - y].
\end{aligned}
\tag{6}
$$

In the proposed method, to impose key point constraint, key point loss is introduced. Theoretically speaking, key points indicate the shapes and directions of faces. If the landmarks of key points are close to those of frontal faces, the transformed faces are also close to frontal faces. Therefore, key points of facial images are extracted. With them, in the training process, we try to minimize the differences of key point positions between the transformed images and the target frontal images. On the other hand, similar to the idea of cycle consistency loss, we also minimize the backward differences. In this way, $\mathcal{L}_{KP}$ is defined as:

$$
\begin{aligned}
\mathcal{L}_{KP} = \ &\mathbb{E}_{x \sim p_{data}(x)}[\| G(x) - y \|] \\
&+\mathbb{E}_{y \sim p_{data}(y)}[\| \overline{G}(y) - x \|].
\end{aligned}
\tag{7}
$$

There are many existing methods to extract facial key points. Among them, methods based on deep learning such as DCNN [20], Dlib [21], TCDCN [22], MTCNN [23], TCNN [24], DAN [25],

and FAN [26] are popular recently. DCNN, Dlib, and FAN are able to extract 68 key points, and we finally employ a face alignment network (FAN) to obtain the landmarks of facial key points. The FAN is constructed based on the hourglass (HG) network for human pose estimation [27]. Actually, face alignment tries to estimate the face pose, and frontal face synthesis transforms face poses to the front. In particular, the method in [27] proposed to stack four HG networks to construct the network and its structure is shown in Figure 2. We follow the settings in the proposed method.

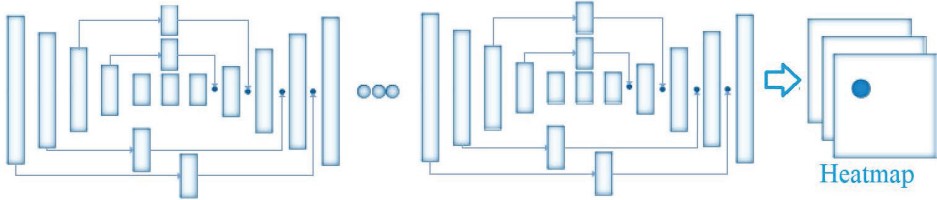

**Figure 2.** The face alignment network (FAN) constructed by stacking four hourglass (HG) networks.

### 2.3. Pose-Invariant Face Recognition

### 2.3.1 Network Architecture

FT-GAN uses a similar network architecture to the one in CycleGAN [19], which consists of a generator network and a discriminator network. The generator network consists of two stride-2 convolutional layers, 9 residual blocks, and two stride-12 fractionally-strided convolutional layers. The discriminator network consists of PatchGAN [28] and the Adam solver [29]. PatchGAN is used to classify whether a $70 \times 70$ patch in an image is real or fake, while the Adam solver is used to compute the $L2$ distance between the transferred key points and input key points. We trained FT-GAN with 50 epochs.

### 2.3.2 Implementation Details

Pose-invariant face recognition with FT-GAN is shown in Algorithm 1. First, we extract the key points of facial images independently. Then, the key points are used in the key-point term of the proposed loss function. Finally, the key point loss term is added and the model of CycleGAN is trained with the facial images. This model is used for frontal face transformation. For face recognition, the proposed framework is compatible with any existing methods, such as Dlib and FaceNet [30]. All the face images were resized to $160 \times 160$.

---

**Algorithm 1** Details of Pose-invariant face recognition with FT-GAN.

---

**Input:** Unconstrained face images for training $X$, frontal face images for training $Y$, unconstrained
     face images for testing $\overline{X}$
**Output:** Face recognition results $\overline{Z}$
 1: Detect faces in $X$, $Y$, and $\overline{Y}$
 2: Extract key points of $X$, $Y$, and $\overline{Y}$
 3: Train FT-GAN with $X$, $Y$, and the corresponding key points according to Equation (3)
 4: Map unconstrained face images $\overline{X}$ to frontal face images $\overline{Y}$ with the corresponding key points
 5: Obtain $\overline{Z}$ with $\overline{Y}$ and face recognition methods (Dlib or FaceNet)
 6: **return** Face recognition results

---

## 3. Experimental Results

### 3.1. Datasets and Settings

In the experiments, we used MultiPIE, which is a large dataset containing more than 750,000 images for face recognition with various poses, illumination, and expression [31]. It is

available via a license from Carnegie Mellon University for internal research purposes (http://www.cs.cmu.edu/afs/cs/project/PIE/MultiPie/Multi-Pie/Home.html). The dataset contains faces captured under different angles, such as $15°$, $30°$, $45°$, $60°$, $75°$, and $90°$. We usedd them to evaluate the performance of both face pose transformation and pose-invariant face recognition. In the evaluations, we randomly chose 2000 images as the training set and the remaining 1000 images as the testing set, which means that we chose 400 in each degree for training and 200 for testing. The training images were further divided into two parts, frontal images and profile images, to obtain the face transformation model. Since the case of $15°$ is rather simple, we only show the results of angles larger than $15°$. In addition, we used the Head Pose Image Database (HPID) to emphasize the performance of pose-invariant face recognition [32], which can be used for any purpose (http://www-prima.inrialpes.fr/perso/Gourier/Faces/HPDatabase.html). The head pose database is a benchmark of 2790 monocular face images of 15 persons with variations of pan and tilt angles from $-90°$ to $+90°$. For every person, 2 groups of 93 images (93 different poses) are available. We only used the frontal images and profile images. As a result, 13 images were used for each group and the diversity of angles was exactly the same as Multi-Pie. The first group containing 1 frontal face and 12 profile faces was used to train the model. Then, the second group was used for testing. We used various measurements for performance in different scenarios and they are introduced in the following. This process was repeated 20 times and average performance was computed.

In the experiments, the proposed method was compared with CR-GAN, which is the state-of-the-art method to achieve frontal face synthesis and learning complete representations for multi-view generation. It uses a two-pathway framework for self-supervised learning. In this way, pose-invariant representations of facial images can be computed. All the experiments were conducted on a workstation with 1080Ti GPU.

## 3.2. Results of Frontal Face Synthesis

In this part of the experiments, we compared the performance of frontal face synthesis, which is shown in Figure 3. We can see that CR-GAN, CycleGAN, and FT-GAN could transform profiles to fronts. However, in these cases, CR-GAN could not provide reasonable results. In some cases, the results of the original CycleGAN were not satisfactory either. For example, in the case of $45°$, the facial shape was quite different from the original frontal face. In the case of $60°$, the symmetrical characteristic was not kept by CycleGAN. These problems can be fixed by the key point alignment of the proposed methods. Therefore, the proposed FT-GAN achieved better frontal face synthesis performance. Besides, for the 1000 testing images, it took about 61 milliseconds to transform an image for CR-GAN, while it took about 38 ms for FT-GAN. This indicates the efficiency of the proposed method.

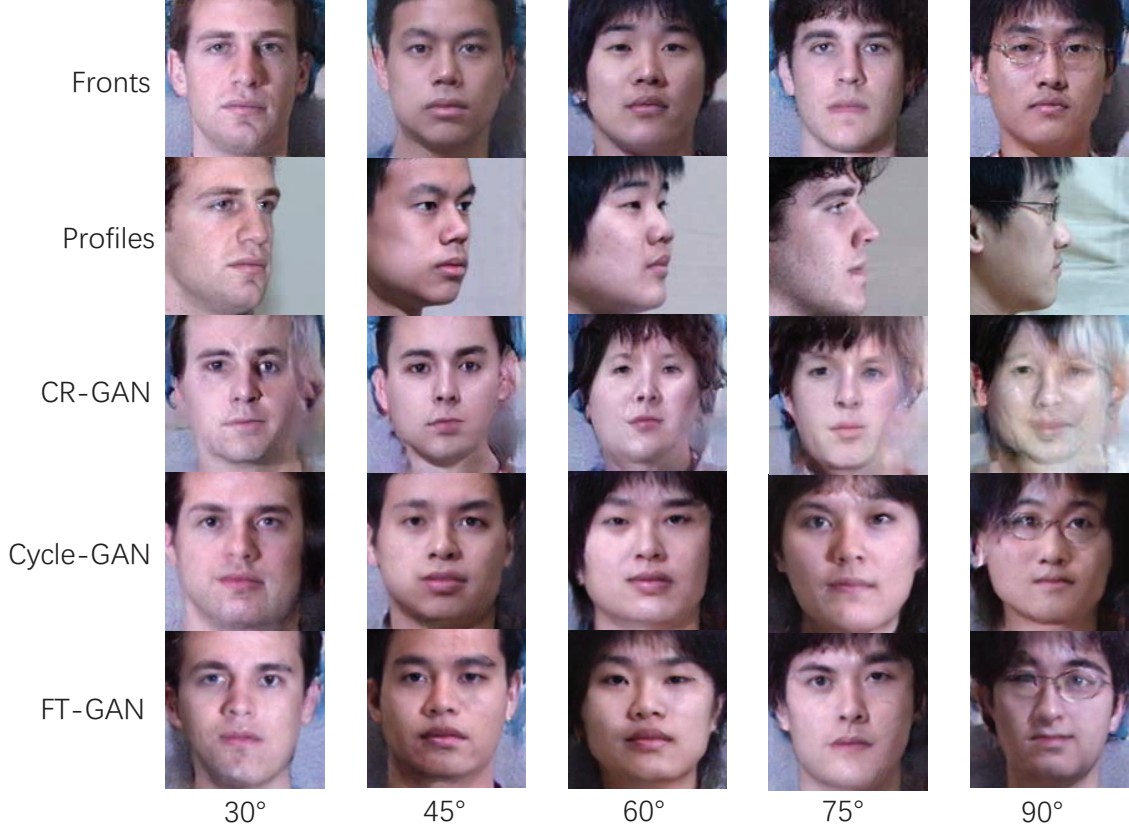

**Figure 3.** Comparison of frontal face synthesis. From top to bottom, we show the original frontal faces, profiles, the results of complete representations with a generative adversarial network (CR-GAN), the original CycleGAN, and face transformation with key points alignment based on generative adversarial networks (FT-GAN). From left to right, we show the results of different facial angles: 30°, 45°, 60°, 75°, and 90°.

## 3.3. Results of Face Recognition with Complex Poses

The state-of-the-art process of face recognition consists of face detection and face verification. Therefore, we looked into the performance in these two steps. As mentioned before, we used two popular face recognition frameworks: Dlib and FaceNet. FaceNet does not contain a face detector. However, it is usually used with MTCNN [23]. Therefore, to evaluate the performance of face detection, MTCNN was used. The results with the original images (origin), transformed images of CR-GAN (CR-GAN), transformed images of CycleGAN (CycleGAN), and transformed images of FT-GAN (FT-GAN) are shown.

First, for face detection, we computed the missing rates, which are defined as the ratios of missed faces in detection. The results on two datasets are shown in Figures 4 and 5. Generally speaking, MTCNN was more robust than Dlib. However, unconstrained face directions influenced the performance of face detection. In this way, larger angles resulted in worse performance. Face transformation computed frontal faces and the performance was improved. Among CR-GAN, CycleGAN, and FT-GAN, the proposed FT-GAN achieved the lowest missing rates on both frameworks.

For face verification, two experiments were conducted: 1:1 matching and 1:$N$ matching. In the 1:1 matching experiment, if the transformed image matched the front image of the same person within the predefined threshold, we denoted it as a successful case. In the 1:$N$ matching experiment, the similarities between a transformed image and all the frontal images were computed. If this transformed image was the most similar to as the frontal image of the same person, we denoted it as a successful case. The accuracy was defined as the ratio of successful matching.

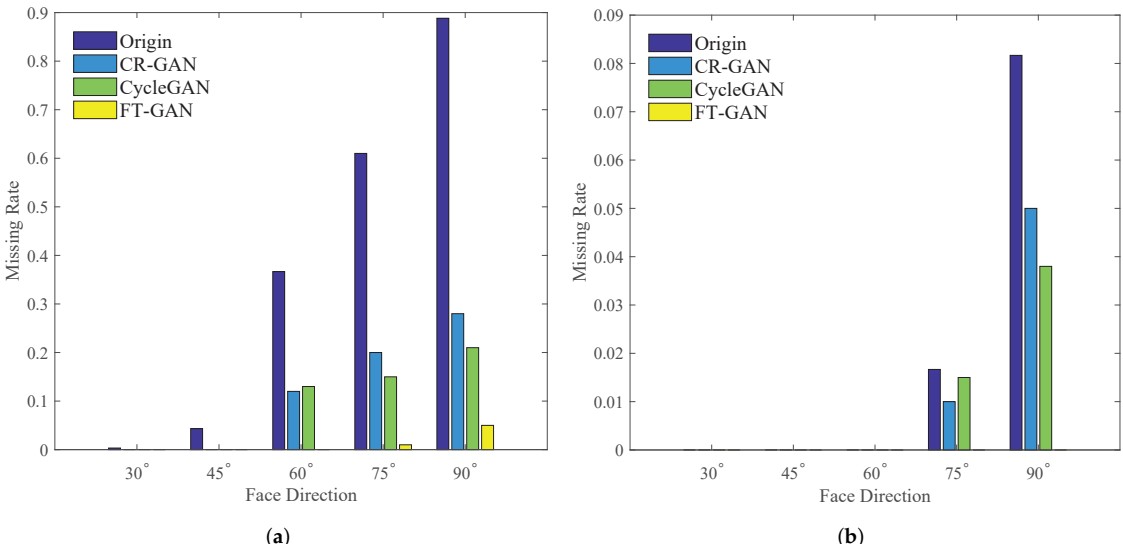

**Figure 4.** The missing rates in face detection for Multi-Pie. (**a**) Dlib; (**b**) MTCNN.

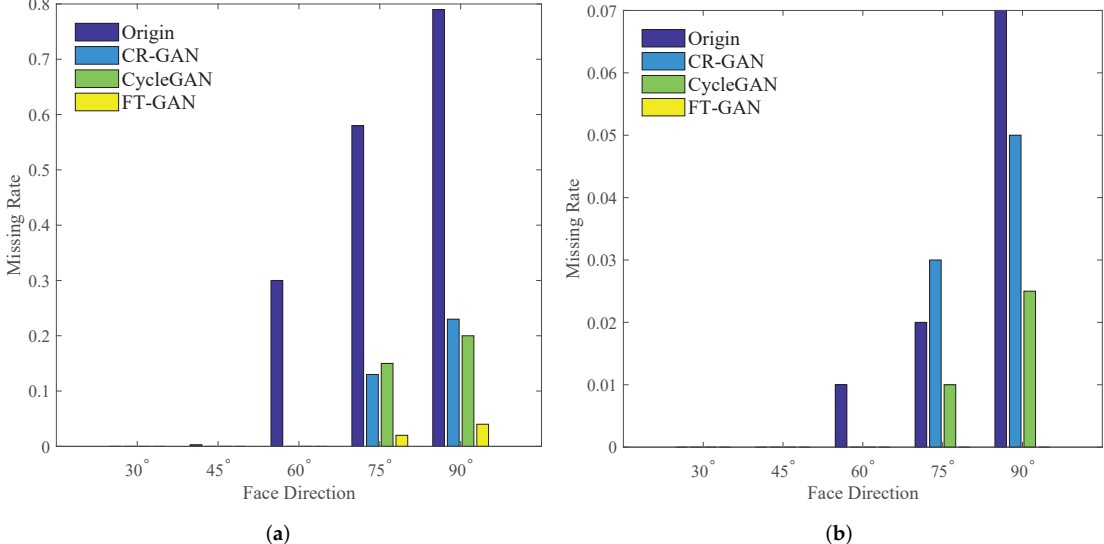

**Figure 5.** The missing rates in face detection for the Head Pose Image Database (HPID). (**a**) Dlib; (**b**) MTCNN.

The results of 1:1 matching for Multi-Pie and HPID are shown in Figures 6 and 7, while the results of 1:*N* matching are shown in Figures 8 and 9. We can see that FaceNet was also better than Dlib in face recognition. Besides, Multi-Pie was more challenging than HPID. However, face transformation did not always provide better performance, especially in the cases of small angles. Current face recognition methods were somehow robust to face direction, and face transformation actually synthesized frontal faces looking different from the original person. With larger angles, such as 60° for Dlib and 75°, the performance using the original images was reduced significantly. In these cases, face transformation was helpful to improve the performance of face matching. We can see that the proposed FT-GAN achieved stable performance under different face directions and it was the best among these three face transformation methods.

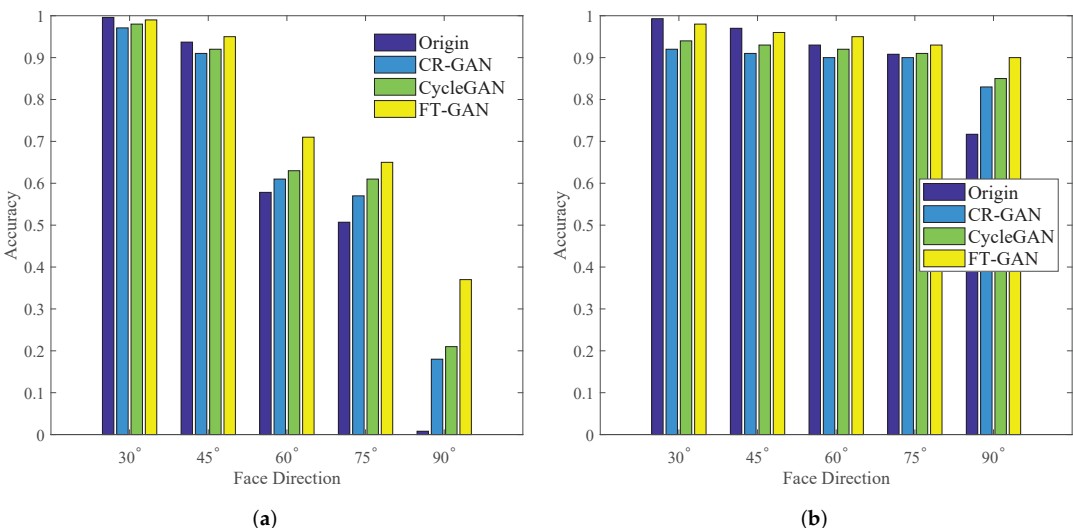

(a)

(b)

**Figure 6.** The accuracy of 1:1 face matching for Multi-Pie. (**a**) Dlib; (**b**) FaceNet.

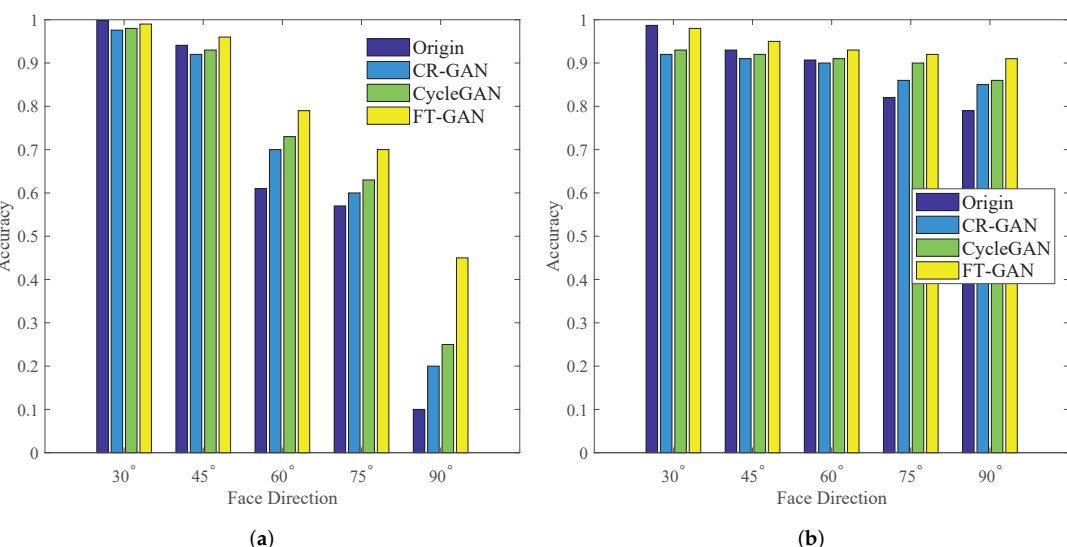

(a)

(b)

**Figure 7.** The accuracy of 1:1 face matching for HPID. (**a**) Dlib; (**b**) FaceNet.

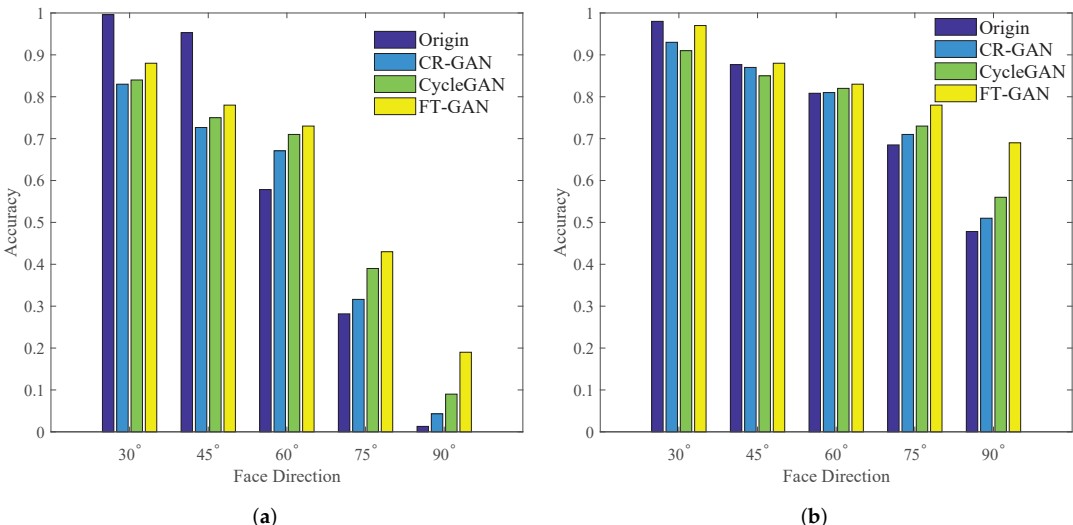

(a)

(b)

**Figure 8.** The accuracy of 1:*N* face matching for Multi-Pie. (**a**) Dlib; (**b**) FaceNet.

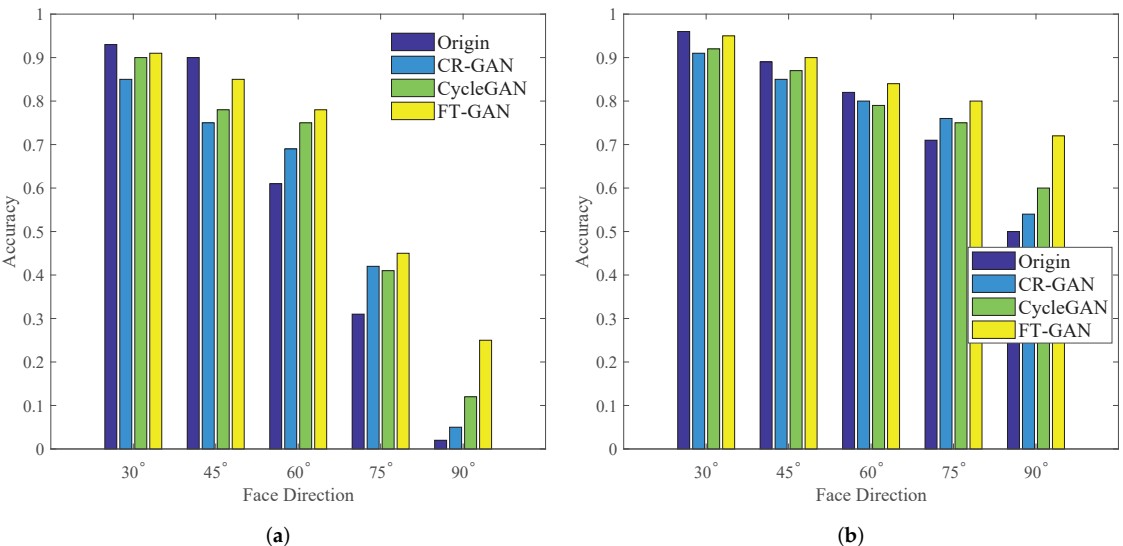

(a)　　　　　　　　　　　　　　(b)

**Figure 9.** The accuracy of 1:*N* face matching for HPID. (**a**) Dlib; (**b**) FaceNet.

### *3.4. Influences of Key Point Extraction*

As mentioned before, there have been several methods for key point extraction. Among them, we chose MTCNN, DCNN, and Dlib to compare with the FAN used in the proposed method. We used FaceNet and the accuracy of 1:*N* face matching on two datasets to demonstrate the advantage of FAN. The results are shown in Figure 10. We can see that their performance was rather close, especially on small facial angles. MTCNN only extracted five key points, which resulted in lower accuracies of the key point loss term. In this way, the performance on large facial angles was worse than the other methods. FAN was the best of all.

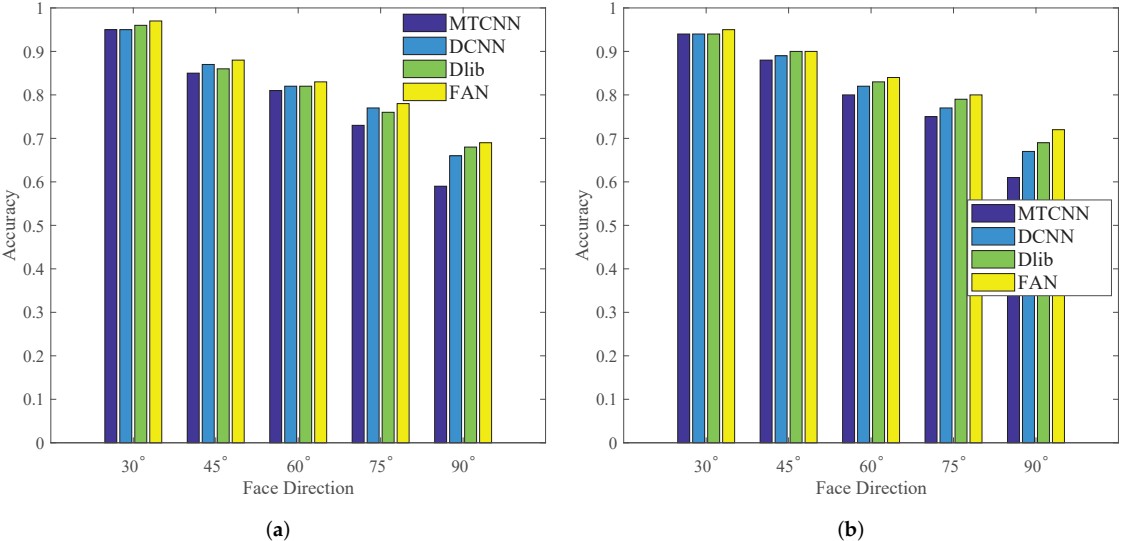

(a)　　　　　　　　　　　　　　(b)

**Figure 10.** The accuracy of 1:*N* face matching with different key point extraction methods. (**a**) Dlib; (**b**) FaceNet.

### 4. Discussion

According to the methodology of face transformation for pose-invariant face recognition and the improvement of experimental performance, the novelty of the proposed FT-GAN can be shown.

First, the proposed method tackles the problem that previous face transformation methods ignore the landmarks of facial key points. We introduce CycleGAN to face transformation. CycleGAN is

used for pixel transformation. In addition, we impose key point alignment to keep the landmarks of facial key points. In this way, an improved loss function is proposed and used in face transformation. Face transformation with key points alignment (FT-GAN) is proposed to synthesize frontal faces and used in pose-invariant face recognition. Therefore, the proposed method is theoretically novel.

Second, comprehensive experiments were conducted and the effectiveness of FT-GAN was indicated. Although the results with face transformation were not always better than the results without face transformation, face transformation provided stable performance under large angles. We compared FT-GAN with CR-GAN in terms of face transformation. FT-GAN outperformed CR-GAN in face transformation, face detection, and face verification. Besides, the experimental comparison demonstrated that the key point extraction method used in the proposed method is a good choice. Two popular face recognition frameworks were employed and the results on both of them emphasize the overall improvement of the proposed method.

## 5. Conclusions

In this paper, a two-task process for face transformation is proposed which consists of pixel transformation with CycleGAN and key point alignment. Compared with previous methods, the proposed method keeps the landmarks of facial key points and details of shapes. In this way, the performance is improved. FT-GAN is used for frontal face synthesis and pose-invariant face recognition. Performance comparison indicates the superiority of FT-GAN over state-of-the-art methods.

**Author Contributions:** Writing original draft, W.Z.; Methodology, L.C.; Formal analysis, C.H.; Data curation, Y.L.; Funding acquisition, K.W.

**Funding:** This work was funded by the National Natural Science Foundation of China (61622205 and 61836002), the Fujian Provincial Natural Science Foundation of China (2018J01573), the Fujian Provincial High School Natural Science Foundation of China(JZ160472), and the Foundation of Fujian Educational Committee (JAT160357).

**Conflicts of Interest:** The authors declare no conflict of interest.

## Abbreviations

The following abbreviations are used in this manuscript:

GAN    Generative Adversarial Networks
CCA    Canonical Correlation Analysis
LDA    Linear Discriminant Analysis
DNN    Deep Neural Network

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
