# Peer review of "FT-GAN: Face Transformation with Key Points Alignment for Pose-Invariant Face Recognition"

_electronics, doi:10.3390/electronics8070807_

Round 1
Reviewer 1 Report
This paper considers a problem of facial recognition methods. Although there have been many studies in the literature, the method used in this paper is rather significant. The paper extracts some key and missing information about the face recognition. The authors also have investigated the problem by using the comprehensive experiments.
So, I would like to recomendthe paper for acceptance.
Author Response
Thanks for the reviewer’s positive support. We conduct careful proofreading to further improve the quality of the manuscript.

Reviewer 2 Report
The paper describes a framework for Face frontalization. The reference application, as also reported in the title of the paper, is the recognition.
A DN approach based on two main components is exploited. The first component, i.e. a CycleGan implementation, works at pixel level. Then, a second step is based on facial landmark alignment.
Experimental results on the MultiPie dataset shown impressive results. However, the paper should be improved a lot from the technical point of view, in order to clarify several issues.
1) The overall architecture of the system is not clear. The network components are coupled and trained end-to-end? In particular, it is not clear the landmark computation and alignment step.
From Figure 1, each block seems to be independent. However, the loss of Eq.1 includes the key-point term.
2) It looks like Figure 1 is not real (not composed by real results). First of all, I suggest to include in the schema the computational blocks, leaving the figures as visual example of the input and output of each block. Moreover, I assumed the keypoint alignment is done after the CycleGan step by applying a spatial transformation. However, the figure contains a different background at the right of the face. This is very strange.
3) Figure 2 is taken from [19]. Please, cite the source. In addition, change the text at line 126: we stack four -> the method in [19] proposes to stack four...
4) Table 1 and Algorithm 1 can be converted into text, or they should be more detailed than the actual ones.
5) Page 4 contains a general formulation of CycleGan plus specific author changes, but it is difficult to recover the novelty proposed in this paper. If the authors think that CycleGan is not sufficiently well-known, please move the standard part to a separate section or an appendix.
6) Experimental evaluation. The dataset is composed by 750000 faces. It is not clear why the authors used 3000 samples only. Moreover, it is not clear how the split into training, validation and testing has been done. Does the test set contain images of the same person of the training? This is crucial also from a theoretical point of view. The system is able to recontruct the non-visible part of the face (e.g. the other side for 90° angles) based on the visible part or based on a training image of the same person? Since the framework uses generative networks, the network could "create" a non-visible part of the face "inspired by" elements of the training set. Thus, the generated face results in a mixture of parts from different people. The Adversarial part of the GAN guarantees a realistic result, but the identity of the person is changed. As a result, the obtained face is no more valid for recognition tasks. This issue is partially resolved if the training set contains the same subject, but the method lacks in portability.
7) FaceNet, as proposed in [23], does not contain a face detector, but only an embedding for face recognition. Figure 4 is not right.
Author Response
Thanks for the reviewer's comments. We have carefully improved the quality of the manuscript according to them. Please refer to the uploaded file.

Reviewer 3 Report
The idea of combining pixel-to-pixel transformation (CycleGAN) and key point alignment is interesting. However, the details of calculating key point alignment loss is unclear. The first paragraph of the introduction was not robust without scientific evidence. There are no detailed explanations if the authors trained CycleGAN or used pre-trained CycleGAN without any changes. Although all ground truth key points are provided in the Multi-PIE, the performance of the key point extraction is important as the key points are not available in the most applications. As there are many methods for facial key point extraction, it can be useful to apply different extraction methods and compare them. For experiments, using only one dataset (Multi-PIE) is not sufficient to show the effectiveness of the proposed method. Also, the random selection of training and test set makes the proposed method less reproducible
Author Response

(The authors gave the same response as above.)

Round 2
Reviewer 3 Report
In this revised paper, the reviewer's concerns and comments were sufficiently addressed and updated. The authors made the paper more robust with presenting different facial keypoint extraction methods and experiments with HPID dataset. I do not have any more concerns.